# An Analysis of the Impact of Annotation Errors on the Accuracy of Deep Learning for Cell Segmentation

**Şerban Vădineanu**[1]                                       S.VADINEANU@LIACS.LEIDENUNIV.NL

**Daniël M. Pelt**[1]                                         D.M.PELT@LIACS.LEIDENUNIV.NL

**Oleh Dzyubachyk**[2]                                        O.DZYUBACHYK@LUMC.NL

**K. Joost Batenburg**[1]                                     K.J.BATENBURG@LIACS.LEIDENUNIV.NL

[1] *Leiden Institute of Advanced Computer Science, Leiden University, Leiden, the Netherlands*

[2] *Leiden University Medical Center, Leiden, the Netherlands*

**Editors:** Under Review for MIDL 2022

## Abstract

Recent studies have shown that there can be high inter- and intra-observer variability when creating annotations for biomedical image segmentation. To mitigate the effects of manual annotation variability when training machine learning algorithms, various methods have been developed. However, little work has been done on actually assessing the impact of annotation errors on machine learning-based segmentation. For the task of cell segmentation, our work aims to bridge this gap by providing a thorough analysis of three types of potential annotation errors. We tackle the limitation of previous studies that lack a golden standard ground truth by performing our analysis on two synthetically-generated data sets with perfect labels, while also validating our observations on manually-labeled data. Moreover, we discuss the influence of the annotation errors on the results of three different network architectures: UNet, SegNet, and MSD. We find that UNet shows the overall best robustness for all data sets on two categories of errors, especially when the severity of the error is low, while MSD generalizes well even when a large proportion of the cell labels is missing during training. Moreover, we observe that special care should be taken to avoid wrongly labeling large objects when the target cells have small footprints.

**Keywords:** Deep learning, cell segmentation, noisy labels

## 1. Introduction

Image segmentation, i.e., the labeling of relevant features in images, has been an important topic for the computer vision community (Guo et al., 2018). In recent years, the use of deep convolutional neural networks for image segmentation has become increasingly popular (Ajmal et al., 2018). Although such algorithms are able to achieve similar performance to human annotators on certain tasks (Esteva et al., 2017), they are heavily dependent on both the quantity and the quality of the training data. The importance of quality is especially prominent in the context of segmentation, where the annotation process is time-consuming and often requires domain-expert knowledge (e.g., in medical imaging). One important issue that arises is the high variability between expert annotators when segmenting anatomical structures from medical images (Benchoufi et al., 2020; Liu et al., 2019; Zhang et al., 2020). For instance, the segmentation of multiple sclerosis poses difficulties to many experts since the lesion area can vary in size, shape or location (Zhang et al., 2019), inducing high inter- and intra-observer variability (Carass et al., 2017). Also, there can be a considerable amount

of disagreement between experts when defining the segmentation border of optic nerve head in retinal images (Edupuganti et al., 2018). These annotation dissimilarities can mean that the manually annotated labels used for segmentation may deviate from the ground truth, which can negatively impact the accuracy of the supervised machine learning models.

In order to compensate for such inconsistencies, various label fusion techniques, e.g., STAPLE (Warfield et al., 2004), VoteNet (Ding et al., 2019), have been proposed to extract an approximation of the ground truth from multi-expert annotations. However, such methods often require multiple opinions for the same data, a process that is costly and slow. In addition to the effort that the research community is putting into alleviating the label inconsistency issue, it is also important to study the actual impact that such label imperfections are causing to the segmentation algorithms. The benefits of such a study are twofold. Firstly, the engineers who use existing deep learning solutions when developing tools would learn whether they can reduce the expert time on annotations by admitting lower quality labels and still achieving the desired results. Secondly, the developers of deep learning techniques can be provided with insights indicating ways to design more robust algorithms with respect to annotation errors.

While the literature proposes multiple methods to mitigate the effect of annotation errors in image segmentation, there are few works evaluating the concrete implications of these errors. In particular, Zlateski et al. (2018) develop a measurement of label quality in the context of semantic segmentation of synthetic urban street view scenes. They apply various levels of simplifications to the segmentation masks of the scene and use a modified version of FusionNet (Quan et al., 2021) and FCN16 (Shelhamer et al., 2017) to generate the predictions. Their results emphasized the need for a large set of coarsely annotated images rather than strongly controlling the label quality. However, the study assumes immediate availability of a large pool of unannotated images with an inexpensive coarse annotation process, which is often impossible to achieve in medical imaging, where even creating coarse labels requires a certain extent of expertise. Heller et al. (2018) emulated three types of perturbations on a liver segmentation data set (Bilic et al., 2019). The errors included the application of random offsets, shifts and flips of pixel labels applied to the annotation images, while the evaluation was performed for UNet (Ronneberger et al., 2015), SegNet (Badrinarayanan et al., 2017) and FCN32 (Shelhamer et al., 2017). The selection of errors was further diversified by Vorontsov and Kadoury (2021) with their work on a MRI brain tumor data set (Henry et al., 2020). They made use of elastic transformations, random crops of the tumor area, constant shifts and random permutations between slices and their labels. Consequently, they observed the effects of the perturbations for multiple learning paradigms based on a UNet backbone. Both studies introduce errors presenting plausible occurrence scenarios. However, each of them is performed on a single manually-annotated data set, whose labels can already be subjected to the errors the authors try to model.

In this paper, we extend previous works by introducing three error-emulation techniques applied to three different data sets. So far, the current annotation error studies on biomedical images have been focusing on segmentation tasks of unitary objects (e.g., organs, tumors). Such objects are limiting most error emulation approaches to create perturbations only at the border of the object's label. We deviate from this approach by proposing an analysis of sparsely distributed objects in the context of cell segmentation. This enables us to not only induce errors at the border of the objects, but also to emulate errors concerning

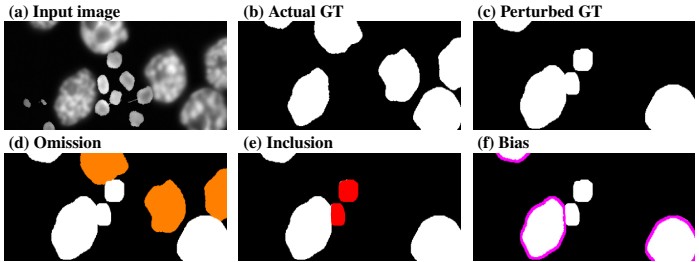

Figure 1: Example of our proposed perturbations. Figure (**a**) shows an input image where the HL60 cells are the target objects and the granulocytes form the background objects. The unmodified ground truth is shown in (**b**), and its perturbed version in (**c**). The errors are highlighted in (**d**) – omission/orange, (**e**) – inclusion/red and (**f**) – bias/purple.

entire regions, such as the complete removal or addition of cells. In addition, we address the shortcomings of using manual annotations as ground truth by employing two perfectly-annotated synthetic data sets of HL60 and granulocytes (Svoboda et al., 2009) and validate our observations on manually-annotated microscopy images (Verma et al., 2020). Moreover, we expand the current analysis by incorporating a network whose architecture diverges from the usual encoder-decoder paradigm.

## 2. Background and methodology

Our analysis is focused on the segmentation task of 2-dimensional vector-valued (e.g., RGB) images, denoted as arrays of pixels $x \in \mathbb{R}^{N \times M \times C}$, where $N$, $M$, $C$ represent the number of rows, columns and channels, respectively. The aim is to find a mapping from $x$ to an output $y \in \mathbb{Z}^{N \times M}$ that subdivides the image into disjoint sets of pixels, each set corresponding to a certain category. In our work, we address the problem of binary cell segmentation by separating only one class of objects from the background. Suppose the image $x$ contains $E$ cells. For each cell $i$ we define the cell label $l^i$ as the binary image in which the pixels belonging to that cell are set to one and all other pixels set to zero:

$$l_{nm}^i = \begin{cases} 1, & \text{if } x_{nm} \text{ belongs to cell } i \\ 0, & \text{otherwise} \end{cases} \quad \forall \; 1 \leq n \leq N, \; 1 \leq m \leq M. \tag{1}$$

Given the set of all cell labels $\mathcal{L} = \{l^1, l^2, \ldots, l^E\}$, a target image for training can be constructed by:

$$y = \sum_{l \in \mathcal{L}} l \tag{2}$$

In order to approximate the desired mapping, we employ convolutional neural networks (CNNs) by passing the input image through a series of successive operations, called layers. The networks are given a set of input images $X = \{x_1, x_2, \ldots, x_{N_t}\}$ and the predicted

output $\hat{Y} = \{\hat{y}_1, \hat{y}_2, \ldots, \hat{y}_{N_t}\}$ is compared against the target output $Y = \{y_1, y_2, \ldots, y_{N_t}\}$ with the goal of minimizing a loss function.

**Annotation errors.** As the true output usually comes from manual annotation, it becomes subjected to human errors, which can hinder the training of CNNs. We model such inconsistencies and separate them into three categories (shown in Figure 1) as follows:

*Omission Errors.* Typical stained tissue scans can include tens or even hundreds of cells of different shapes and sizes (Diem et al., 2015). When creating segmentation masks for such diverse and populated images, it is possible that an expert annotator may unintentionally ignore a certain proportion of the relevant cells. We call such absence of cell annotations *omission errors* and we develop a systematic method of altering the ground truth mask by removing a ratio of the present cells from the label set $\mathcal{L}$. An example of cell removal is showcased in Figure 1(d). We define $\mathcal{L}^S \subseteq \mathcal{L}$ as a random subset of size $S \leq E$, where $S$ is chosen to satisfy the omission rate $r_\omega = \frac{S}{E}$. The label after omission is comprised of the binary labels corresponding to the remaining cells $y^\omega = \sum_{l \in \mathcal{L} \setminus \mathcal{L}^S} l$.

*Inclusion Errors.* Another issue that can arise in tissue scans is the accidental annotation of cells belonging to the wrong category. In such cases, an annotator might sometimes include some fundamentally different cells due to their proximity or apparent resemblance to the correct cells. We incorporate this *inclusion error* into our analysis with various amounts of severity, which correspond to the amount of "wrong" cells that we choose to include into the label set. One such case is presented in Figure 1(e). We define $\Lambda = \{\lambda^1, \lambda^2, \ldots, \lambda^F\}$ as a set of binary labels for other objects within $x$ and $\Lambda^S \subseteq \Lambda$ as a random subset of size $S \leq F$, where $S$ is chosen to satisfy the inclusion rate $r_\phi = \frac{S}{F}$. The resulted subset is then added to the label set $\mathcal{L}$ before creating the final label $y^\phi = \sum_{l \in \mathcal{L} \cup \Lambda^S} l$.

*Bias Errors.* Another important factor that deserves attention is the ambiguity that is often present when delimiting the cell borders. Often, it is difficult for annotators to precisely distinguish the true outline of cells. This can lead to annotations that deviate from the gold standard (ground truth), inducing *bias* into the data. Such biases can manifest in the form of creating cell labels that excessively cover the actual cell surface, as can be observed in Figure 1(f). Moreover, the opposite can also happen, when the annotator "shrinks" the corresponding label relative to the true area of the cell. We consider both cases in our study and we also control the amount of bias we introduce by expanding and reducing the sizes of the cell labels that are present in our data sets. In order to model the annotation bias, we employ morphological operations (Serra, 1983). Specifically, we simulate excessively covering cells by applying a dilation operation $\oplus$ to the target image $y$ a number of $q$ times, where $q$ is randomly chosen between 1 and $q_{max}$. Similarly, we simulate the shrinking of cells by applying an erosion operation $\ominus$ $q$ times, where $q$ is randomly chosen between 1 and $q_{max}$.

## 3. Experiments

### 3.1. Experimental Setup

In this work, we considered three types of convolutional neural networks based on their wide adoption and distinctive characteristics. Our selected networks include, with the chosen configurations further detailed in Appendix A:

- **UNet** (Ronneberger et al., 2015) – encoder/decoder architecture, decoder with transposed convolutions, direct connections between the encoder and decoder;

- **SegNet** (Badrinarayanan et al., 2017) – encoder/decoder architecture, decoder with unpooling, no connections between the encoder and decoder;

- **Mixed-scale dense network** (**MSD**; Pelt and Sethian (2017)) – densely connected architecture, dilated convolutions.

We performed our experiments using PyTorch (Paszke et al., 2019) implementations of our chosen network architectures, while keeping their structure, e.g., number of layers, similar to their original implementation. For our two-class segmentation problem the true output will be a two-channel image, where a pixel on the first channel is 1 if it corresponds to a pixel of the background and 0 otherwise, while the reverse is true for the second channel. A soft-max activation is used on the output of the final layer, while all intermediate layers are paired with a ReLU function. We aim to minimize the Dice loss by using ADAM optimizer (Kingma and Ba, 2015) while training the network for 20 epochs on the synthetically-generated data and for 50 on the manually-annotated images, the latter epoch count being larger due to the increased complexity of the images. After each epoch, the model is tested on a validation set selected as a separate portion of 30% from the training data and the model with the lowest validation score is kept. Our qualitative metric is the Sørensen–Dice coefficient, which we compute for the entire test set and average over 10 runs. Moreover, whenever a network reaches an untrainable state, i.e., it only segments the background, we discard the model and restart training with a different initialization. The performance comparisons were validated using Wilcoxon tests (Rey and Neuhäuser, 2011).

### 3.2. Synthetic Data

These experiments were conducted on simulated microscopy images of HL60 nuclei cells and granulocytes (Svoboda et al., 2009). The images obtained from the Masaryk University Cell Image Collection[1] were generated by a virtual microscope (Wiesner et al., 2019). An image-label sample pair for each data set is shown in Figure 2. The different size and position distribution of the two cell types make them good candidates for our analysis since the same generated perturbation can affect them differently. This will enable us to apply our observations to a broader variety of cells. For each category of cells, the data set consists of 30 volumes, each volume being separated into 129 slices of $565 \times 807$ 16-bit pixels. We used 25 volumes for training, while 5 were kept for testing. We selected the slices that had a non-empty label, resulting in an average number of 84 slices per volume. For these data sets we assume having one annotator per volume, thus, we emulate the errors once per volume.

**Omission errors.** We perform the omission for $r_\omega \in \{10\%, 20\%, 30\%, 50\%, 70\%\}$. The results presented in Figures 2(e,h) show that this category of errors has limited impact on the networks' performance when we consider moderate cases ($r_\omega \leq 30\%$). MSD and UNet show a similar robust behavior to moderate omissions, while SegNet presents a pronounced downward trend, with a 10% reduction in Dice score for 30% omission, relative to no omission. For relatively large omissions ($r_\omega > 30\%$), MSD maintains a relatively low

---

1. https://cbia.fi.muni.cz/datasets

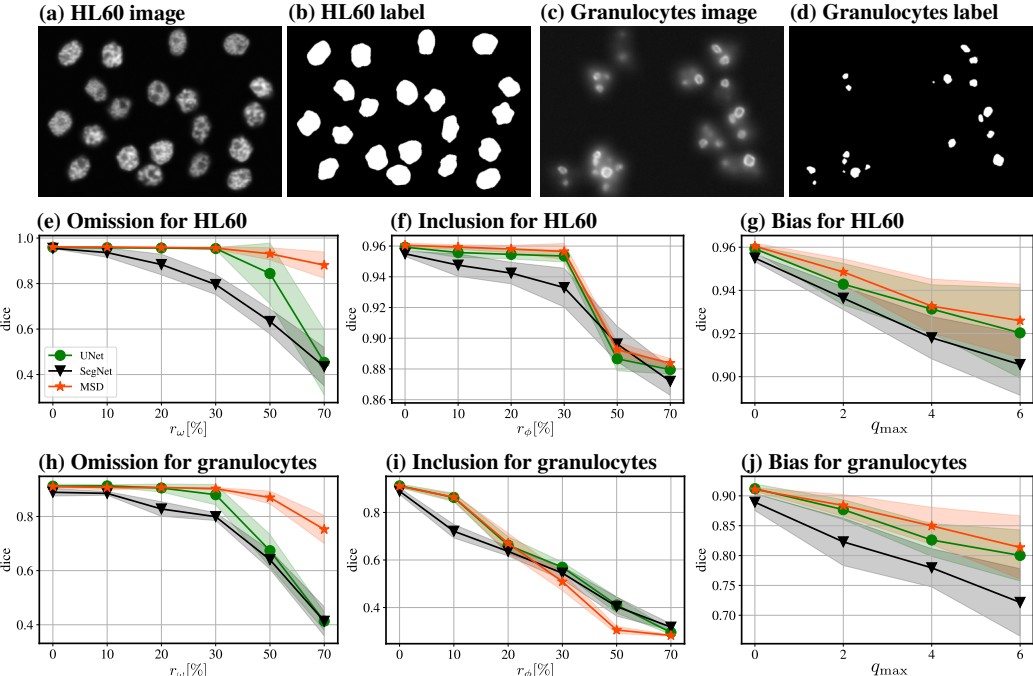

Figure 2: Example images and results for the synthetic data sets. Figures (**a**,**b**) and (**c**,**d**) show image/label pairs of simulated microscopy slices. (**e**–**j**) show the Dice score of trained networks on the test set as a function of perturbation severity, for HL60 cells (**e**–**g**) and granulocytes (**h**–**j**). Results are shown for: omission (**e**,**h**), inclusion (**f**,**i**), and bias errors (**g**,**j**). The shade around the curves corresponds to the standard deviation of the results.

reduction in performance even for $r_\omega = 70\%$. However, this comes with the caveat that, for omissions above 30%, the training process of MSD occasionally collapses to an untrainable state. Both the training instability and the limited reduction in accuracy for large omission rates of MSD are a consequence of its design. The low number of parameters required by MSD might enable it to be less prone to overfitting on the wrongly labeled data, but also to become less stable when the label quality is substantially deteriorated.

**Inclusion errors.** In order to add inclusion perturbations, we merge the volumes of the HL60 nuclei cells and the granulocytes, while defining one data set to be the main one, with its labels being $\mathcal{L}$, while the other one becomes secondary, with its labels being $\Lambda$. In our experiments, we have $r_\phi \in \{10\%, 20\%, 30\%, 50\%, 70\%\}$. Figures 2(f,i) illustrate that the inclusion perturbation results in different behaviors for the models, depending on the data set it is applied to. In the case of HL60, SegNet presents a slow decreasing trend until $r_\phi = 30\%$, while UNet and MSD appear to be unaffected by the moderate inclusion. Also, since the granulocytes occupy a much smaller area in each slice than the HL60 cells, their addition into the latter's volume does not heavily impact the models even for 70% inclusion, resulting in a loss in performance of less than 1%. Moreover, Figure 2(i) shows that wrongly including

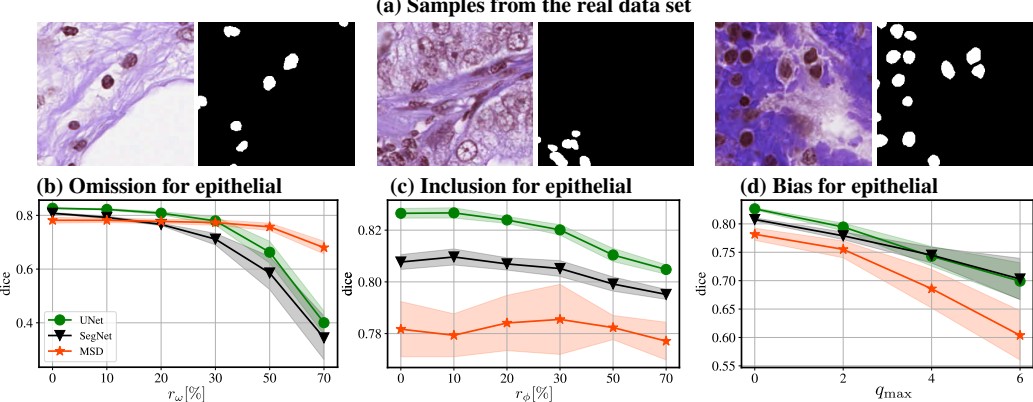

Figure 3: Example images and results for the manually-annotated data set. Figure (**a**) shows image/label pairs of stained-tissue images. Figures (**b**–**d**) show the Dice score of trained networks on the test set as a function of perturbation severity. Results are shown for: omission (**b**), inclusion (**c**), and bias errors (**d**). The shade around the curves corresponds to the standard deviation of the results.

large objects into the segmentation mask severely impacts all networks' capabilities, with an average Dice score drop of 23% for every 10% increase in $r_\phi$.

**Bias errors.** For this type of perturbation, we performed our analysis by choosing $q_{max}$ from $\{2, 4, 6\}$. Figures 2(g,j) show that introducing label bias through random morphological operations creates a descending trend for all network architectures. However, this trend presents different magnitudes depending on the data set. In the case of HL60 cells we observe a decrease in Dice score of up to 5%, while for granulocytes the performance drops to 19%. This decline is a consequence of the much smaller footprint of the granulocytes' labels in relation to the background. Thus, mistakes on the outline of smaller cells are more costly than for their larger counterpart. In addition, we notice here that MSD and UNet perform similarly on the synthetic images, with SegNet lagging behind, by 6% and 10% on average for HL60 (Figure 2(g)) and granulocytes (Figure 2(j)), respectively.

### 3.3. Manually-annotated Data

Following the observations drawn from the synthetic data we aim to extend them to a segmentation task of manually-annotated stained tissue images. We selected the data set belonging to MoNuSAC 2020 challenge (Verma et al., 2020), which contains H&E stained tissue images belonging to multiple organs. The data were gathered with the purpose of performing automatic cell segmentation, which can provide crucial information about the organ's health. This data set is comprised of 310 8-bit images of various sizes containing four types of cells: epithelial, lymphocytes, macrophages and neutrophils. Among these types, we selected the epithelial cells to be the target of our task, while considering the rest as background. The selection was motivated by the larger presence of the epithelial cells on a both per-image and per-data-set basis. Hence, we are left with 96 images for training

and 37 for testing. Moreover, due to the varying size of the images, an extra preprocessing step was performed. The step involved separating each image into $256 \times 256$ patches using a sliding widow technique, while allowing for an overlap of 64 pixels between patches. A few samples of the selected patches are shown in Figure 3(a). Also, given the variability in size, quality and provenience of the data, we assume an individual annotator for every single image. Thus, we will apply our perturbation framework to each image separately.

The omission and bias-inducing processes are performed similarly to the synthetic data. For inclusion, we choose the main type of cells to be epithelial, while the lymphocytes form the secondary category. We chose lymphocytes since their pairing to epithelial cells is the most prevalent in the data set. We show the experimental results in Figures 3(b–d). In the case of omission, one notable difference from the synthetic data is the slight performance gap between MSD and the UNet/SegNet pair for $r_\omega < 30\%$. Nonetheless, this gap decreases the more error we allow, showing MSD to plateau at 77% Dice score until we remove 50% of the cell labels, while the other networks are severely affected (18% and 26% reduction for UNet and SegNet, respectively). When it comes to inclusion, the segmentation performance, similarly to the HL60 cells, appears to be rather unaffected by wrongly labeled additional cells until 30% inclusion. Moreover, the larger rates ($\geq 50\%$) inflict a more modest loss in the Dice score compared to the HL60 volumes due to the poorer fit the models have on real data. Since their learned parameters may not be a perfect fit on the data, the models can allow small perturbations of their input without suffering large losses. The bias on epithelial cells shows UNet and SegNet to develop an increasing gap from MSD, which reaches a 22% reduction in Dice score for $q_{max} = 6$. Here, MSD appears to suffer from the lack of complexity since this data set presents a more complex background with high variability between images, impeding, thus, a very good distinction of the correct cells. This tendency is further exacerbated by the perturbations applied to the cells' masks. Additional results for both synthetic and manually-annotated data are given in Appendix B.

## 4. Conclusion

Understanding the consequences of labeling errors is of great importance for the field of biomedical image segmentation. Our study provided an insight into meaningful issues that can be present in the annotation process for cell segmentation. We emulated three different labeling errors (omission, inclusion and bias) for two perfectly-labeled synthetic data sets and one manually-annotated data set and observed their impact on the results of three networks. We found that wrongly including large objects into the segmentation labels drastically decreases the quality of the predictions, while smaller objects are filtered out more easily when moderately included ($r_\phi \leq 30\%$). We also observed that, even in low amount, the presence of bias deteriorates the predictions for all cell types, especially for relatively smaller cells such as granulocytes and epithelial cells. Finally, we observed that moderate omissions ($r_\omega \leq 30\%$) present a negligible impact to both MSD an UNet, with the latter slightly outperforming the former on the manually-annotated data set. However, for larger omissions, MSD still retains a competitive Dice score. This robustness to omissions can be exploited in settings where the expert annotator would be required to label just a portion of the present cells, significantly reducing the annotation costs. Also, MSD could be used to pre-process training labels for more complex, but noise-sensitive, learning algorithms.

## Acknowledgments

This research was supported by the SAILS program of Leiden University. DMP is supported by The Netherlands Organisation for Scientific Research (NWO), project number 016.Veni.192.235.

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

**Appendix A.**

In order to provide a more general comparison, we chose the default configurations for the network architectures that we used. Therefore, UNet consists of four downscaling blocks followed by four upscaling blocks, where a downscaling block is composed of two convolutional layers and the upscaling block consist of one transposed convolutional layer. For SegNet, we built the encoder half out of five downscaling blocks, out of which the first two are double convolution blocks and the last three are formed from triple convolutions. The decoder of SegNet mirrors the encoder, hence, it is formed from five upscaling blocks, where the first three are triple convolution blocks and the last two are double convolution blocks. Lastly, MSD also follows its default configuration, namely a depth of 100 layers, a width of one and dilation factors chosen from $\{1, 2, 3, 4, 5, 6, 7, 8, 9, 10\}$.

## Appendix B.

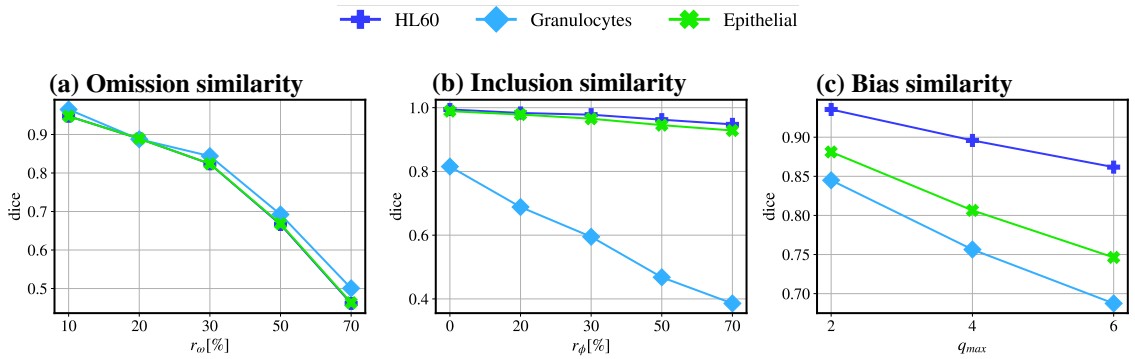

Figure B.1: Similarity measure between perturbations and the original labels. Figures (**a**,**b**,**c**) show the Dice similarity between the labels before and after applying the perturbations. The score is computed over the entire training set and averaged over 10 random initializations for each perturbation.

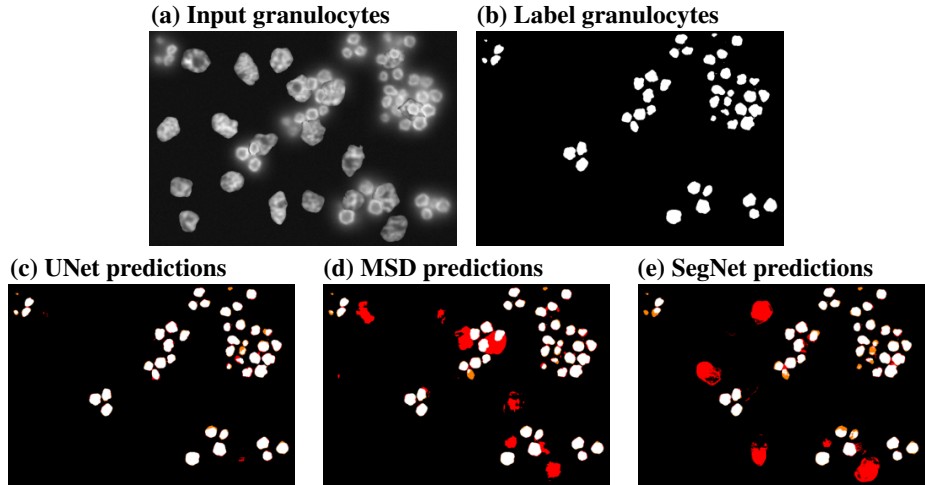

Figure B.2: Network predictions when trained on 30% inclusion. Figures (**a**,**b**) show an image/label pair of granulocytes. The averaged predictions over 10 differently-initialized models are shown in (**c**) for UNet, (**d**) for MSD and (**e**) for SegNet. The red pixels correspond to false positives and the orange pixels correspond to false negatives.

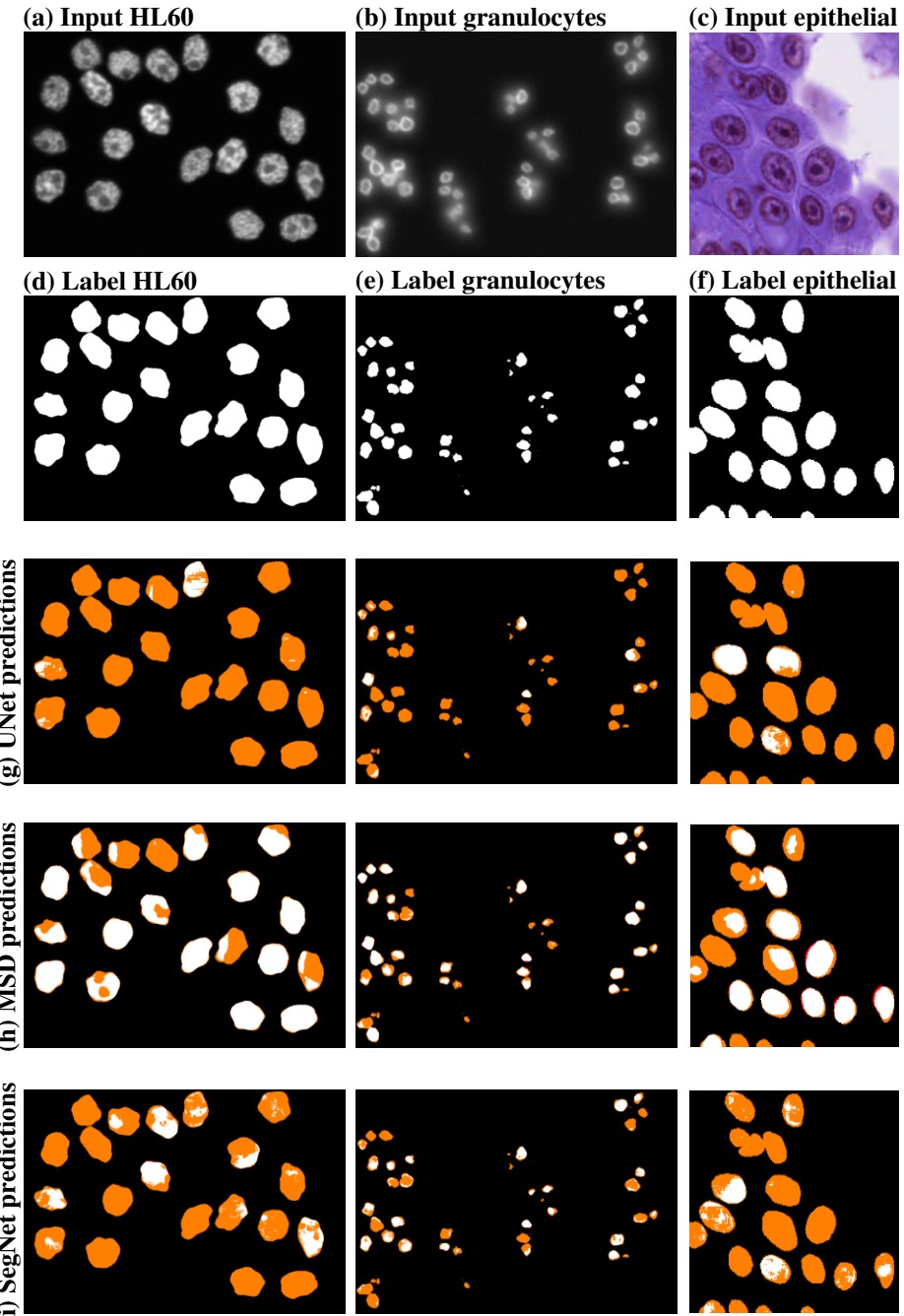

Figure B.3: Network predictions when trained on 70% omission. Figures (**a**,**d**), (**b**,**e**) and (**c**,**f**) show image/label pairs for HL60 cells, granulocytes and epithelial cells, respectively. The averaged predictions over 10 differently-initialized models are shown in (**g**) for UNet, (**h**) for MSD and (**i**) for SegNet. The red pixels correspond to false positives and the orange pixels correspond to false negatives.

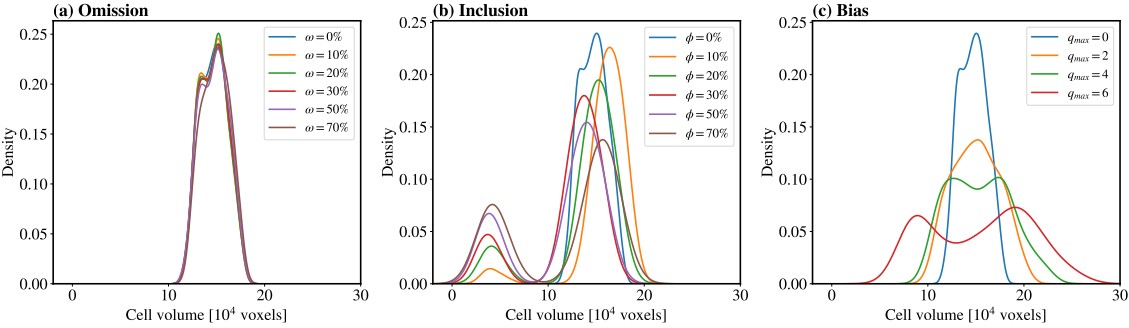

Figure B.4: The influence of our errors on the distribution of cell volumes for the HL60 data set. Figure (**a**) presents the effect of omission, Figure (**b**) of inclusion and Figure (**c**) of bias errors.

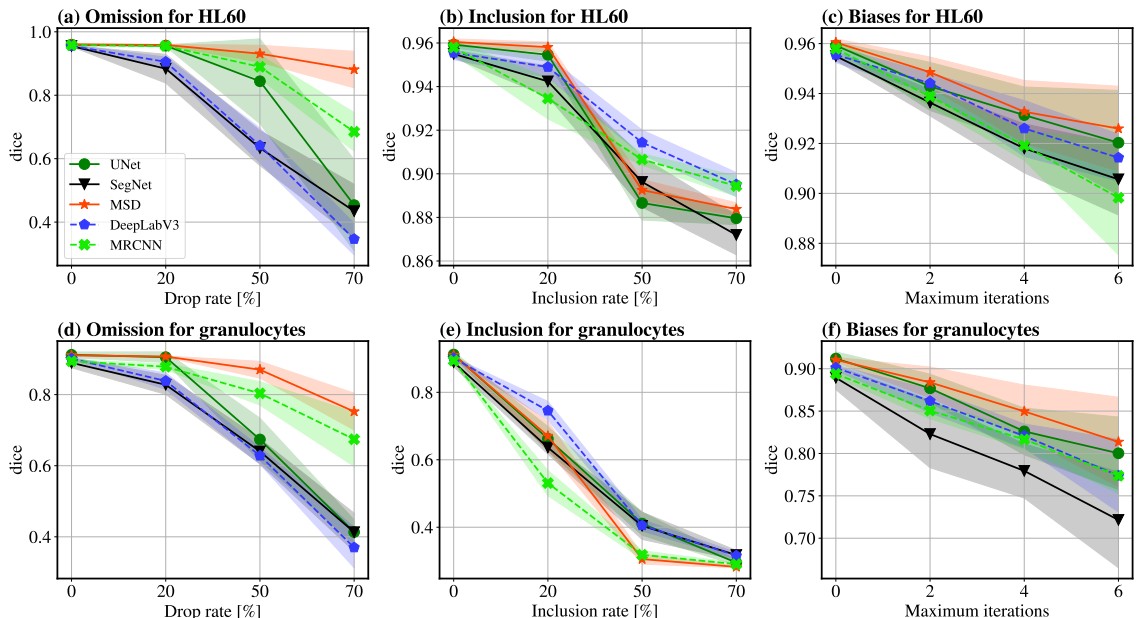

Figure B.5: Results for the synthetic data sets including two additional networks: DeepLabV3 (Chen et al., 2017) and Masked RCNN (He et al., 2017). Figures (**a**–**f**) show the Dice score of trained networks on the test set as a function of perturbation severity, for HL60 cells (**a**–**c**) and granulocytes (**e**–**f**). Results are shown for: omission (**a**,**d**), inclusion (**b**,**e**), and bias errors (**c**,**f**). The shade around the curves corresponds to the standard deviation of the results.

