# OpenReview forum: "An Analysis of the Impact of Annotation Errors on the Accuracy of Deep Learning for Cell Segmentation"
_MIDL.io/2022/Conference — MIDL 2022_

### Official Review · Reviewer_KUJL · 2022-01-21

**Confidence:** 5
**Preliminary Rating:** 4
**Recommendation:** Poster

**Summary:**

This paper assess the impact of annotation errors on machine learning-based segmentation. Application: Cell segmentation. Findings: wrongly including large objects into the segmentation labels drastically decreases the quality of the predictions, while smaller objects are filtered out more easily when moderately included; bias deteriorates the predictions and moderate omissions have a negligible impact.


**Strengths:**

The paper proposed an analysis of sparsely distributed objects in the context of cell segmentation instead of otherwise common  error emulation approaches based on perturbations.

There are no obvious flaws in the scientific approach and the chosen error metrics seem sound.


**Weaknesses:**

- a bit too bold conclusions
- limited to cell segmentation domain
- limited discussion about practical implications

Would it be possible to integrate these findings into segmentation training to make the resulting model as optimal as possible, i.e., a bit like active learning?

What are the practical implications of these findings. Would this be suitable for, e.g., automated quality control of individual labellers?


**Deanonymize Review:**

no

**Detailed Comments:**

would this approach transfer to other domains beyond segmentation of histopathological images?

"wrongly including large objects into the segmentation labels drastically decreases the quality of the predictions, while smaller objects are filtered out more easily when moderately included" seems to be obvious, if including more bias towards wrong information that the networks will make larger errors.

probably "moderate omissions present a negligible impact" has been shown with the many papers about weakly supervised segmentation, e.g.,
https://www.cv-foundation.org/openaccess/content_iccv_2015/html/Pathak_Constrained_Convolutional_Neural_ICCV_2015_paper.html
https://ieeexplore.ieee.org/abstract/document/8698843?casa_token=Cj3LJzm1vPcAAAAA:KPnl4PgbtDcIoAnrcNRsjP5jOIGbwkSP-7XV9TLBzvQvbuPYW1gPkkLBvUikxH4xEssqZ3Sz
https://openaccess.thecvf.com/content_ICCV_2019/html/Shimoda_Self-Supervised_Difference_Detection_for_Weakly-Supervised_Semantic_Segmentation_ICCV_2019_paper.html
https://www.sciencedirect.com/science/article/pii/S1361841518306145?casa_token=tXgT5asTvocAAAAA:VOZ2I2pYF78jU3AkpEGGgBFuR11Ml2nLCaIIWxZhuhWEONYbjoUuMY6E0382dya35mYoq3qT


minors:
golden standard -> Gold standard (https://en.wikipedia.org/wiki/Gold_standard_(test))


**Final Rating After The Rebuttal:**

4: Weak Accept

**Justification Of The Final Rating:**

Thank you for addressing the points raised above and adding clarifications to the presented work. I have no further comments.
Thank you for addressing the points raised above and adding clarifications to the presented work. I have no further comments.

**Paper Type:**

validation/application paper

**Questions To Address In The Rebuttal:**

Would it be possible to integrate these findings into segmentation training to make the resulting model as optimal as possible, i.e., a bit like active learning?

What are the practical implications of these findings. Would this be suitable for, e.g., automated quality control of individual labellers?


**Special Issue:**

no

---

### Meta-Review · Area_Chair_aSdT · 2022-02-20

**Recommendation:** Accept (Poster)
**Confidence:** 5

**Metareview:**

The authors present a systematic analysis of the effects of different annotations errors (inclusion, omission, bias) on the behaviors of 3 different deep learning architectures for segmentation. While the analysis is focused on the cell segmentation in microscope images, the reviewers agree the proposed evaluation framework is applicable to many other segmentation challenges in medical images. We see no obvious flaws in the scientific approach, the chosen error metrics seem sensible and the presented results are comprehensive and interesting to the community.

---

### Decision · Program_Chairs · 2022-02-28

Accept